# *Ocimum* Species: A Review on Chemical Constituents and Antibacterial Activity

**DOI:** 10.3390/molecules27196350

**Published:** 2022-09-26

**Authors:** Hendra Dian Adhita Dharsono, Salsabila Aqila Putri, Dikdik Kurnia, Dudi Dudi, Mieke Hemiawati Satari

**Affiliations:** 1Department of Conservative Dentistry, Faculty of Dentistry, Universitas Padjadjaran, Sumedang 45363, West Java, Indonesia; 2Department of Chemistry, Faculty of Mathematics and Natural Sciences, Universitas Padjadjaran, Sumedang 45363, West Java, Indonesia; 3Department of Livestock Production, Faculty of Animal Husbandry, Universitas Padjadjaran, Sumedang 45363, West Java, Indonesia; 4Department of Oral Biology, Faculty of Dentistry, Universitas Padjadjaran, Sumedang 45363, West Java, Indonesia

**Keywords:** *Ocimum* species, chemical constituents, antibacterial

## Abstract

Infection by bacteria is one of the main problems in health. The use of commercial antibiotics is still one of the treatments to overcome these problems. However, high levels of consumption lead to antibiotic resistance. Several types of antibiotics have been reported to experience resistance. One solution that can be given is the use of natural antibacterial products. There have been many studies reporting the potential antibacterial activity of the *Ocimum* plant. *Ocimum* is known to be one of the medicinal plants that have been used traditionally by local people. This plant contains components of secondary metabolites such as phenolics, flavonoids, steroids, terpenoids, and alkaloids. Therefore, in this paper, we will discuss five types of *Ocimum* species, namely *O. americanum*, *O. basilicum*, *O. gratissimum*, *O. campechianum*, and *O. sanctum*. The five species are known to contain many chemical constituents and have good antibacterial activity against several pathogenic bacteria.

## 1. Introduction

Infection by microbes is one of the main problems that causes several diseases. One of the causes of infection is bacteria that can have an impact on public health. Based on the collection of data from 52 sentinel hospitals in North America for 7 years (1998–2004) on contemporary strains of 12,737 strains of pediatric patients under 18 years of age, *E. coli* as a pediatric pathogen ranks in the top six [1]. Mapanguy et al. [2] also reported that oral antibiotics such as cefixime, amoxicillin, and ciprofloxacin were resistant to *E. coli* infection about 50–60%. In addition, microorganisms can also cause wound infections and inhibit healing. Some of the bacteria associated with wound infections are *Staphylococcus aureus*, *Escherichia coli*, *Pseudomonas aeruginosa*, *Klebsiella pneumoniae*, *Streptococcus pyogenes*, *Proteus* spp., *Streptococcus* spp., and *Enterococcus* spp. [3,4]. Bacteria is also can be pathogenic in the skin, such as *Staphylacoccus*, *Micrococcus*, and *Corynebacterium* sp. [5]. In relation to dental and oral health, *Streptococcus sanguinis* and *Streptococcus mutans* can cause dental caries [6]. 

Some of the infections mentioned can be treated with antibiotics. However, several studies have proven that antibiotics can cause resistance [7]. Antibiotic resistance occurs due to the use of drugs in large quantities, causing selection pressures on human and natural microbial systems. Microbes can undergo mutations to survive, thereby reducing antibiotic sensitivity [8,9]. Infections due to drug resistance have caused the death of up to 700,000 people every year worldwide [10]. There are several antibiotic resistances, namely vancomycin and teicolanin against *S. aureus* [11], gentamicin aminoglycoside against *E. faecalis* and *E. faecium* with percent resistance of 30% to 50% [12], penicillin against *S. pneumonia* [13], and *E. coli* caused resistance of fluorokuinolon more than 80% and gentamicin more than 40% [12]. 

Therefore, the use of natural antibacterial ingredients is one solution that has great potential. *Ocimum* species are herbal plants that are available in Indonesia. *Ocimum* species are native to tropical areas such as southern Asia, Africa, and India [14]. *Ocimum* comes from the *Lamiaceae* family, which has about 50 to 150 species [15]. Due to its pharmacological effects, this plant has been widely used traditionally for the treatment of headaches, coughs, diarrhea, constipation, warts, and kidney damage [16]. These properties come from the secondary metabolite components that are abundant in *Ocimum* plants such as steroids, tannins, alkaloids, flavonoids, and phenolics [17]. In addition, the abundant components of essential oils make *Ocimum* a plant that can fight the growth of organisms [18,19]. Therefore, in this paper, we will discuss five species of *Ocimum* that have been tested in vitro to have antibacterial activity against several Gram-positive and Gram-negative bacteria.

## 2. Chemical Constituents and Antibacterial Activity of *Ocimum* Species

### 2.1. Ocimum americanum

*O. americanum* is native to Africa and is 15 to 60 cm tall with sub-rectangular striated branches [20,21]. The leaf shape is intact or faintly serrated, lanceolate ellipse, glandular spots, and glabrous. The color of flower is pink, white, or purplish with an elongated circle shape. The fruits are small, pitted notelets, and mucilaginous [21]. It is commonly known as hoary basil or mosquito plant and has three chemotypes, namely spicy, camphoraceous, and floral-lemony [22] Traditionally, this plant is used for the treatment of digestive, respiratory, and sedative disorders. It also has benefits as a cough medicine, treating bronchitis, immune disorders, relieving toothache, and dysentery, which is commonly used orally [20,23]. The extract of *O. americanum* was also used for tobacco flavoring, tea, and body fragrance. The leaves and branches were used for insecticides against mosquitoes, bees, flies, and other insects. In Africa, the Swahili tribe utilizes the aerial parts of the plant for the treatment of high blood pressure and stomach aches [24]. In addition, local people in the Tamil Nadu area use a decoction of the leaves as a medicine for diabetes, constipation, diarrhea, hemorrhoids, and dysentery [25].

*O. americanum* has several phytochemical components, such as alkaloids, flavonoids, phenolic, tannins, terpenoids, saponin, steroids and glicosides. Some studies reported that saponin, phenolic, and tannins are found in less polar solvent such as ethyl acetate leave extract [24], while glicosides and steroids are commonly found in methanol extract [25]. Moreover, phenolic, flavonoids, saponin, and tannins were found in the aqueous extract of leaves and flowers [26]. The pharmacological activities found in *O. americanum* are antioxidant, antifungal, antimicrobial, anti-insecticide and larvicide, and gastric cytoprotective antiulcer effect [27,28]. Chemical compounds in *O. americanum* can be seen in Table 1, Appendix A.

### 2.2. Ocimum basilicum

*Ocimum basilicum*, commonly called sweet basil, is one of the species of genus *Ocimum* from Asia, Africa, and South America regions [36,37]. *O. basilicum* can live in different climates and ecology, grows in cool humid areas to tropical areas with the temperatures between 6 and 24 °C, and also favors warm conditions [38]. This plant is the species of *Ocimum* which is commercially available in the market [39]. It has six different morphologies, namely true basil with green leaves, small-leaf basil, which belongs to green cultivars with short narrow leaves and grows rounded, lettuce-leaf with broad leaves, purple basil A, which has green leaves with purple flowers and stems, purple basil B, where the leaves, flower, and stems are purple, and purple basil C, which has a similarity to purple basil B and has broad listered leaves [40,41]. *O. basilicum* is 20 to 80 cm tall with glabrous and woody stems, large green leaves, and is broadly elliptical, measuring 2.5 to 5 cm × 1 to 2.5 cm. The flowers are red, pink, or white, with a size of 3 mm, and are arranged in terminal spikes [42].

Traditionally, the fruit of *O. basilicum* was used as folk medicine against inflammation, diarrhea, worm infestation, and eye-related disease [43]. Leaves and flowers of *O. basilicum* were used as tonic and vermifuge, and can also be used as a tea to treat nausea, flatulence, and dysentery. *O. basilicum* contains essential oils that are commonly used to treat colds, seizures, and treatment of wasp stings and snakebites [44]. The polysaccharide component of *O. basilicum* was traditionally used as cancer treatment in China [45,46]. In South Europe, they used *O. basilicum* as Mediterranean food, such as the cuisines of Italian and Greek [47]. 

*O. basilicum* contains the main components that are beneficial for health such as calcium, phosphorus, vitamin A, vitamin C, and beta carotene [48]. Phytochemical constituents contained in *O. basilicum* are alkaloids, flavonoids, phenols, saponins, tannins, terpenoids, carbohydrates, cardiac glycosides, cholesterol, glycosides, and phlobatannis [49,50]. Therefore, it has the potential to have anti-inflammatory, antimicrobial, antivirals, anticancer, antifungal, antidiabetic, anti-allergic, analgesic, cardioprotective, and immunomodulatory properties [50,51,52,53]. Then, flavonoids and phenolic compounds gave antioxidant activity [54,55]. Chemical compounds in *O. basilicum* can be seen in Table 2, Appendix A.

### 2.3. Ocimum gratissimum

*O. gratissimum*, with the common name of clove basil, is a species of family *Labiate* which grows in tropical region, namely India and West Africa [71,72]. It is 1–3 m tall and has leaves that are 3–4 cm × 1–2 cm [73]. The flowers have several colors, such as yellowish white, greenish purple, hairy, calyx greenish purple, brown seeds, and not slimy [74]. In Africa, eastern, central, and western Kenya, *O. gratissimum* is commonly found in scrub and disturbed highland forests at elevations of 600 to 2400 m above sea level [75]. 

Traditionally, *O. gratissimum* was used for the treatment of cough, fever, snakebites, mosquito repellent, anemia, inflammation, and diarrhea [75,76]. It has several bioactivities, namely antioxidant, anti-inflammation, antimycotoxicogenic, antibacterial, antifungal, antimalaria, and antiseptic activities [77,78,79,80]. The phytochemical components of *O. gratissimum* are alkaloids, saponins, tannins, phlobatannins, glycosides, phenols, anthraquinones, flavonoids, and terpenoids [81,82]. Some of the chemical compounds contained in *O. gratissimum* are listed in Table 3, Appendix A.

### 2.4. Ocimum campechianum

*O. campechianum* is a plant of the Lamiaceae family group which originates from the tropics of South and Central America. This plant is commonly known as “Albahaca de campo” or “Albahaca silvestre”, used by local people for traditional medicine or culinary purposes [95,96]. This plant has a height of 1 m and contains essential oil components with two types of aromatic leaves, namely glandular trichomes, peltate, and capitate. In addition, *O. campechianum* contains components of flavonoids, polyphenols, and tannins [97,98]. Traditionally, this plant is used as a decoction of leaves, ointments, and for the treatment of fever, cough, bronchitis, diarrhea, dysentery, and hypertension. In addition, *O. campechianum* can also be used as an emmenagogue that helps childbirth [99]. In terms of pharmacological effects, this plant extract is known to have antifungal, antioxidant, antiradical, antiproliferative, and analgesic activities [96,100,101]. In addition, it also has potential as a natural larvicide and pesticide [102]. The data of chemical constituents in *O. campechianum* can be seen in Table 4, Appendix A.

### 2.5. Ocimum sanctum

*O. sanctum* is a plant from the Lamiaceae family, commonly known as basil or tulsi, which is native to India and is widely distributed as a cultivated plant throughout Southeast Asia [106]. It is known as a sacred plant in India and a symbol of purity. It got the name of “Tulasi” from Tulasi Devi, one of the Lord Krishna’s eternal consorts. Tulasi was a Gopi who was said to have fallen in love with Krishna and was cursed by his wife Radha. He is very similar to Vishnu [107]. In India, it is used for religious plants and important events such as weddings [108]. *O. sanctum* is 30–75 cm tall with an herbaceous shape, and is erect, more-branched, and hairy-soft [107]. It has pointed or blunt leaves, the leaves are oval, and the flowers are tightly coiled with a pale red or dark red color [109]. 

Traditionally, *O. sanctum* was used for the treatment of diarrhea, chronic fever, malaria, skin disease, bronchitis, dysentery, insect bite, arthritis, and bronchial asthma [110,111]. *O. sanctum* has several bioactivities such as anticancer, antispasmodic, antifertility, antimicrobial, anti-inflammatory, antioxidant, antifungal, analgesic, antidiabetic, cardio protective, adaptogenic, antiemetic, and hepatoprotective [112,113,114,115]. It has several phytochemicals, namely terpenoids, phenolic, flavonoids, glycosides, and propenyl phenols [116]. Moreover, it also contains vitamin C, A, and minerals such as zinc, iron, and calcium [117]. The protein content in *O. sanctum* is 4.2 g, then 0.5 g fat, 25 mg carbohydrates, 287 mg phosphorus, 25 mg calcium, vitamin C per 100 g, and 15.1 mg iron [118]. The chemical constituents included in *O. sanctum* are listed in Table 5, Appendix A.

## 3. Antibacterial Activity of *Ocimum* Species

Tests of antibacterial activity against *Ocimum* species have been carried out on both Gram-positive and Gram-negative bacteria. Based on the data in Table 6, Table 7 and Table 8, the *Ocimum* extracts that were widely tested and gave relatively high activity were essential oil extracts. In general, the five *Ocimum* species have good antibacterial activity. However, based on the inhibition zone data in Table 6, it can be seen that *O. americanum*, *O. basilicum*, and *O. sanctum* had higher activity than the other two species, where the diameter of the inhibition zone reached 20 to 40 mm. In Gram-positive bacteria, *O. americanum*, *O. basilicum*, and *O. sanctum* gave the highest inhibition zone against *S. aureus*, while *O. gratissimum* gave the highest inhibition zone against *E. faecalis*, and *O. campechianum* gave the highest inhibition against *L. ivanovii*. Furthermore, in Gram-negative bacteria, *O. americanum* gave the highest inhibition zone against *P. gingivalis* and *P. intermedia*, *O. basilicum* against *P. aeruginosa*, *O. gratissimum* against *P. mirabilis*, *O. campechianum* against *E. coli*, and *O. sanctum* against *Yersinia enterocolitica.*

Furthermore, MIC is an antibacterial test used to determine its inhibitory activity. Based on de Aguiar et al. [157], MIC concentrations in the range 101–500 µg/mL have strong activity and in the range 500–1000 µg/mL have moderate activity. As for Table 7, testing of *O. basilicum* against several bacteria, such as *S. mutans*, *S. aureus*, *B. cereus*, *L. monocytogenes*, *E. coli*, *P. aeruginosa*, and *S. typhi*, showed MIC values below 100 µg/mL, which means that they provide very strong activity, as well as in *O. sanctum*. The other three *Ocimum* species only showed strong to weak activity.

**Table 7 molecules-27-06350-t007:** Data of minimum inhibitory concentration (MIC) of *Ocimum* species.

Microorganism	Extract	MIC (µg/mL)					Positive Control (µg/mL)	Reference
		1	2	3	4	5		
**Gram-Positive Bacteria**								
*S. mutans*	Essential oil	0.04% *v/v*	18	-	-	-	-	[126,129,158]
	Lauric acid	-	156	-	-	-	-	
	β-Sitosterol	-	25,000	-	-	-	-	
*L. casei*	Essential oil	0.04% *v/v*	-	-	-	-	-	[126]
*E. faecalis*	Essential oil	500	-	-	-	-	-	[41,129]
	β-Sitosterol	-	25,000	-	-	-	-	
*E. faecium*	Essential oil	500	-	-	-	-	-	[41]
*S. aureus*	Essential oil	200	-	1000	-	2.50		[41,55,132,133,134,135,137,147,159]
	70% hydroethanolic	1840	-	-	-	-	-	
	Ethanol	-	-	-	-	4280	^5e^6	
	Hexane	-	-	-	-	2.30	-	
	Linalool	-	32	-	-	-	-	
	Methanol	-	-	-	>2000	-	-	
	Rosmarinic acid	-	-	-	>2000	-	-	
	Eugenol	-	-	-	1000	-	-	
	Caryophyllene	-	-	-	-	50	-	
*B. cereus*	70% hydroethanolic	1540	-	-	-	-	-	[132,137,159]
	Essential oil	-	18–36	-	-	-	-	
	Eugenol	-	-	-	-	25	-	
*S. epidermidis*	Essential oil	300	-	-	-	-	-	[41,145]
	Bark	-	500	-	-	-	-	
*S. phyogenes*	Essential oil	-	50	-	-	-	-	[139,140]
	Ethanol	-	-	7	-	-	-	
*Listeria monocytogenes*	Essential oil	-	36	-	-	-	-	[41,153,160]
	Ethanol	-	-	2150	-	-	-	
	Hydroethanolic	-	10,000	-	-	-	^2b^ > 150 ^2b^ < 150	
*B. subtilis*	Methanol	-	625	-	-	-	-	[143]
*S. sanguinis*	Lauric acid	-	78	-	-	-	-	[158,161]
	Nevadensin	-	3750	-	-	-	-	
*S. faecalis*	Ethanol	-	-	125	-	-	-	[144]
**Gram-Negative Bacteria**								
*P. gingivalis*	Essential oil	350	-	-	-	-	-	[146]
*P. intermedia*	Essential oil	350	-	-	-	-	-	
*F. nucleatum*	Essential oil	700	-	-	-	-	-	
*P. vulgaris*	Essential oil	400	-	-	-	-	-	[41]
*A. baumannii*	Essential oil	1000	-	-	-	-	-	[151]
*E. coli*	Essential oil	-	9–18	1000	-	2.25	-	[132,133,134,159,162]
	Hexane	-	-	-	-	2.50	-	
	(−)-β-elememe	-	-	-	-	>200	-	
*S. typhimirium*	Methanol:DMSO	>200	-	-	-	-	-	[26,139,159,163]
	Essential oil	-	1600	-	-	-	-	
	Leave	-	-	60–250	-	-	-	
	(−)-β-elememe	-	-	-	-	100	-	
*S. dysenteriae*	Bark	-	10,000	-	-	-	-	[145]
*P. aeruginosa*	Essential oil	-	9–18	-	-	>4.50	-	[55,132,162]
	Linalool	-	1024	-	-	-	-	
*P. multocida*	Essential oil	-	2300	-	-	-	-	[154]
*K. pneumoniae*	Hydroethanolic	-	20,000	-	-	-	^2b^10,000 ^2o^ < 7.80	[164]
*M. morganii*	Hydroethanolic	-	10,000	-	-	-	^2o^20,000 ^2o^ < 7.80	[164]
*P. mirabilis*	Hydroethanolic	-	>20,000	-	-	-	^2b^ < 150 ^2o^ < 7.80	[164]
*Coliform bacilli*	Seed	-	-	2500–7000	-	-	-	[165]
*Shigalla* sp.	Ethanol	-	-	40,000	-	-	-	
*Salmonella* sp.	Ethanol	-	-	60,000				
*P. syringae*	Methanol	-	-	-	>2000	-	^4p^2.50	[96]
	Rosmarinic acid	-	-	-	>2000	-	^4p^2.50	
	Eugenol	-	-	-	500	-	^4p^2.50	
*S. flexneri*	(−)-β-elememe	-	-	-	-	100	-	[159]
*S. typhi*	Essential oil	-	9	-	-	-	-	[153,159]
	Methyl eugenol	-	-	-	-	50	-	
*E. herbicola*	Essential oil	-	-	-	-	2 µL/mL	^5k^8	[166]
*P. putida*	Essential oil	-	-	-	-	1 µL/mL	^5k^4	
*Aeromonas hydrophila*	Ethanol	-	-	-	-	3820	^5e^5	[135]
*Vibrio harveyi*	Ethanol	-	-	-	-	4460	^5e^7	
*Vibrio vulnificus*	Ethanol	-	-	-	-	580	^5e^6	

(-) = Test not performed; 1 = *O. americanum*, 2 = *O. basilicum*, 3 = *O. gratissimum*, 4 = *O. campechianum*, 5 = *O. sanctum**;* Positive control: ^b^(Ciprofloxacin), ^e^(Tetracycline), ^k^(Streptomycin), ^o^(Imipenem), ^p^(Cloramphenicol).

The MBC data were used to show the ability of the compound to kill bacterial growth. Based on the MBC data in Table 8, the species from *Ocimum* that had the best activity in killing bacterial growth was *O. sanctum* which could be seen in the MBC values against *B. cereus*, *E. herbicola*, and *P. putida.* Therefore, based on the data of inhibition zone, MIC, and MBC, the *Ocimum* species that have more potential as natural antibacterials are *O. basilicum* and *O. sanctum.*

**Table 8 molecules-27-06350-t008:** Data of minimum bactericidal concentration (MBC) of *Ocimum* species.

Microorganism	Extract	MBC (µg/mL)					Reference
		1	2	3	4	5	
**Gram-Positive Bacteria**							
*S. mutans*	Essential oil	0.08% *v*/*v*	-	-	-	-	[126,129,158]
	Lauric acid	-	2500	-	-	-	
	β-Sitosterol	-	50,000	-	-	-	
*L. casei*	Essential oil	0.30% *v*/*v*	-	-	-	-	[126]
*E. faecalis*	β-Sitosterol	-	50,000	-	-	-	[129]
*S. aureus*	Essential oil	-	-	1000	-	-	[24,55,133,155]
	70% hydroethanolic	7340	-	-	-	-	
	Linalool	-	>1024	-	-	-	
	Caryophyllene	-	-	-	-	>200	
*B. cereus*	70% hydroethanolic	6150	-	-	-	-	[137,159]
	Eugenol	-	-	-	-	50	
*S. phyogenes*	Essential oil	-	100	4.2	-	-	[139,140]
*Listeria monocytogenes*	Essential oil	-	>20,000	-	-	-	[160,164]
	Ethanol	-	-	2150	-	-	
*S. epidermidis*	Bark	-	2000	-	-	-	[145]
*B. subtilis*	Methanol	-	625	-	-	-	[143]
*S. sanguinis*	Lauric acid	-	1250	-	-	-	[158,161]
	Nevadensin	-	15,000	-	-	-	
**Gram-Negative Bacteria**							
*P. gingivalis*	Essential oil	700	-	-	-	-	[146]
*P. intermedia*	Essential oil	700	-	-	-	-	
*F. nucleatum*	Essential oil	1400	-	-	-	-	
*A. baumannii*	Essential oil	8000	-	-	-	-	[151]
*E. coli*	Essential oil	-	-	1000	-	-	[133,159]
	(−)-β-elememe	-	-	-	-	200	
*S. typhimirium*	Essential oil	-	3200	100–300			[139,159,163]
	(−)-β-elememe	-	-	-	-	>200	
*S. dysenteriae*	Bark	-	20,000	-	-	-	[145]
*P. aeruginosa*	Linalool	-	>1024	-	-	-	[55]
*K. pneumoniae*	Hydroethanolic	-	>20,000	-	-	-	[164]
*M. morganii*	Hydroethanolic	-	>20,000	-	-	-	
*P. mirabilis*	Hydroethanolic	-	>20,000	-	-	-	
*S. flexneri*	(−)-β-elememe	-	-	-	-	200	[159]
*S. typhi*	Methyl eugenol	-	-	-	-	100	
*E. herbicola*	Essential oil	-	-	-	-	8	[166]
*P. putida*	Essential oil	-	-	-	-	4	

(-) = Test not performed; 1 = *O. americanum*, 2 = *O. basilicum*, 3 = *O. gratissimum*, 4 = *O. campechianum*, 5 = *O. sanctum*.

## 4. Interaction of Essential Oils to Bacteria

Essential oils usually contain chemical components in the terpenoid and phenylpropanoid groups. The five *Ocimum* species were proven to contain many essential oil components, especially in the monoterpenoid and sesquiterpenoid groups, as well as some phenylpropanoids [167,168]. Therefore, studies on antibacterial activity have also been carried out on the essential oil components. Essential oils of a plants can inhibit both Gram-positive and negative bacteria. However, some studies reported that essential oils were more sensitive to Gram-positive bacteria, but others reported more sensitive to Gram-negative bacteria [169]. Based on El-Shenaway et al. [170], essential oils were more sensitive against Gram-positive bacteria. This was because of differences in cell wall structures. In Gram-negative bacteria, there was an external capsule which prevented penetration of essential oils into microbial cells. The capsule is composed of a more complex bacterial cell wall with a 2–3 nm thick peptidoglycan layer, where there is an outer membrane on the outer layer of peptidoglycan. The presence of the outer membrane is one of the differences between the cell walls of Gram-negative and Gram-positive bacteria. Peptidoglycan and this outer membrane have strong covalent bonds to Braun lipoproteins. This causes the hydrophobic structure of the essential oil to more easily penetrate the cell walls of Gram-positive bacteria [169].

The ability of essential oils to inhibit bacterial growth is due to their hydrophobicity. It increases cell permeability and leak cell constituents [171]. Essential oils will cross cell wall and cytoplasmic membrane which arrange a lot of polysaccharide layers, fatty acids, and phospholipids. As a result, the interaction between lipophilic compounds from essential oils and various structures found in cell walls and membranes causes a cytotoxic effect [172]. Furthermore, essential oils can also agglomerate in the cytoplasm and cause damage to the protein and lipid layers [173].

In the cell membrane, there is a process of ATP production. The action of essential oils can affect changes in intracellular and external ATP balance. This results in a disturbance with ion loss and a decrease in membrane potential, a decrease in the amount of ATP, and a collapse of the proton pump. [174,175]. This will compromise vital functions such as energy systems, synthesis of structural macromolecules, and secretion of enzymes for growth [171]. In addition, essential oils can affect the pH conditions of bacteria. The presence of essential oils on the membrane can disrupt pH homeostasis and cause a significant decrease in pH. Then, the membrane will lose its capacity to block protons. This is because at low pH, the hydroxyl groups in essential oils do not dissociate so that their hydrophobicity will increase, resulting in easier interaction with bacterial cell membrane lipids [176,177].

There are several methods for extracting essential oils from *Ocimum* species, such as hydrodistillation, steam distillation, solvent extraction, enfleurage, cohobation, and maceration [178]. However, this method requires a fairly long process if it is to be consumed simply in the community. As an alternative, the *Ocimum* plant has been widely used traditionally to cure bacterial infections. *Ocimum* can be processed into juice to relieve toothache and treat otitis. This plant can also be made into an infusion for mouthwash. Decoction of this plant can provide an anesthetic effect and act as an antiseptic. Decoction of the leaves and stems can treat diarrhea, fever, and inflammation of the mucous membranes of the nose [179].

## 5. Conclusions

*O. americanum*, *O. basilicum*, *O. gratissimum*, *O. campechianum*, and *O. sanctum* are five types of *Ocimum* species that have abundant chemical components and antibacterial activity. Based on data on the chemical components of the five *Ocimum* species, it is known that the most abundant compounds are terpenoids. *O. basilicum* is the most commonly used and widely available on the market. However, the activity data of the five *Ocimum* species show that this plant can be a natural product that has potential as a natural antibacterial agent.

## Figures and Tables

**Table 1 molecules-27-06350-t001:** The Chemical Constituents of *O. americanum*.

Class of Chemical Constituents	Chemical Constituents	RT (min)	References
Fatty acids	3-Hydroxy-3-methl pentanoic acid (**1**)	4.71	[26,29,30,31]
	Citronellic acid (**2**)	-	
	Stearic acid (**3**)	29.32	
	Linoleic acid (**4**)	30.40	
	α-Linolenic acid (**5**)	31.49	
	Stearidonic acid (**6**)	-	
	Palmitoleic acid (**7**)	26.70	
Fatty alcohols	2-Hexyl-1-decanol (**8**)	21.67	[29]
	Octyl acetate (**9**)	10.48	[29]
Organic acids	Chicoric acid (**10**)	-	[26,31]
Ketones	2-Hydroperoxyheptane (**11**)	5.27	[29,30]
	3-Hepten-2-one (**12**)	6.53	
	Pulegone (**13**)	-	
	Carvone (**14**)	-	
Enone	Mesityl oxide (**15**)	5.52	[29]
Aldehydes	Octanal (**16**)	6.14	[29,30]
	Citral (**17**)	11.79	
	Citronellal (**18**)	5.96	
	Perillaldehyde (**19**)	nd	
Ester	Hexyl acetate (**20**)	6.32	[29]
Organic nitro	Nitrocyclohexane (**21**)	7.05	[29]
Alcohols	1-Octanol (**22**)	7.54	[26,29,30,32]
	Quinic acid (**23**)	-	
	Menthol (**24**)	-	
	4-Carvomenthol (**25**)	-	
	Safrole (**26**)	-	
	Carveol (**27**)	-	
	Verbenol (**28**)	-	
Dialkyldisulfides	2-Methyl-7-octadecyne (**29**)	21.91	[29]
Unsaturated hydrocarbons	9-Eicosyne (**30**)	22.14	[29]
Phenolic	Vanillin (**31**)	6.72	[26,30]
	Ellagic acid (**32**)	-	
	Eugenol (**33**)	20.36	
Monoterpenoids	α-Thujone (**34**)	-	[29,30,33,34]
	Neral (**35**)	11.19	
	Geranyl acetate (**36**)	21.3	
	Linalool (**37**)	7.41	
	β-Myrcene (**38**)	-	
	Geraniol (**39**)	5.76	
	Linalyl acetate (**40**)	-	
	α-Pinene (**41**)	4.87	
	Fenchone (**42**)	-	
	Verbenone (**43**)	-	
	Verbenol (**44**)	-	
	*p*-Cymene (**45**)	-	
	α-Terpinene (**46**)	-	
	α-Phellandrene (**47**)	-	
	β-Phellandrene (**48**)	-	
	Limonene (**49**)	10.60%	
	Carvacrol (**50**)	-	[29,30,33,34]
	Borneol (**51**)	-	
	Thymol (**52**)	-	
	1-Menthone (**53**)	-	
Diterpenoids	Phytol (**54**)	24.80	[29]
	Phytene-2 (**55**)	21.51	
Sesquiterpenoids	β-Bisabolene (**56**)	16.45	[29]
	Humulene (**57**)	15.54	
	(*E*)-β-Famesene (**58**)	-	
Triterpenoids	Verbascoside (**59**)	-	[26]
Phenylpropanoids	Cinnamic acid (**60**)	-	[26,30]
	Fertaric acid (**61**)	-	
Flavonoids	Vicenin-2 (**62**)	-	[26,35]
	Eriodictyol-7-*O*-glucoside (**63**)	-	
	Vitexin (**64**)	-	
	Rutin (**65**)	4.55	
	Genkwanin (**66**)	-	
	Dihydroxy-tetramethoxy(iso)flavone (**67**)	-	
	Cirsilineol (**68**)	-	
	Cirsimaritin (**69**)	-	
	Pilosin (**70**)	-	
Alloxazines and isoalloxazines	Riboflavin (**71**)	-	[26]

(-) = Test not performed; RT = Retention Time; RI = Retention Index.

**Table 2 molecules-27-06350-t002:** The chemical constituents of *O. basilicum*.

Class of Chemical Constituents	Chemical Constituents	RT (min)	RI	References
Carboxylic acid ester	Cyclohexyl formate (**72**)	7.30	1304	[56,57]
Aliphatic aldehydes	Citral (**16**)	6.57	-	[56,58]
Aliphatic alcohol	3-Octanol (**73**)	11.64	-	[59]
Amines	Phenylethanolamine (**74**)	8.77	-	[56,57]
Phenols	2,3,5-Trimethylphenol (**75**)	7.10	-	[56,58]
	Eugenol (**33**)	7.19	-	
Fatty alcohols	4-Hexen-1-ol acetate (**76**)	857	4.428	[56,58,60,61]
	1-Octen-3-ol (**77**)	10.78	979	
Pyrans	2,3-Dehydro-1,8-cineole (**78**)	11.18	-	[57]
Organoheterocyclic compounds	*cis*-Linalool oxide (**79**)	-	1070	[56,57,58]
	3-Methyl-2-phenylindole (**80**)	12.73	1710	
Phenylpropanoids	*trans*-4-Methoxycinnamaldehyde (**81**)	8.57	-	[56,57,60,62]
	Methyl cinnamate (**82**)	-	1338	
Cycloalkenes	Methyl ethyl cyclopentene (**83**)	5.63	-	[56,58]
Monoterpenoids	α-Terpinol (**84**)	20.16	-	[56,57,58,60,62,63,64]
	1-Menthone (**49**)	5.76	-	
	Levomenthol (**85**)	5.91	-	
	Nerol (**86**)	-	1229	
	Neral (**34**)	-	1249	
	Geraniol (**38**)	-	1256	
	Citral (**16**)	6.57	1270	
	β-Myrcene (**37**)	11.38	-	
	*p*-Menth-3-ene (**87**)	6.74	977.5	
	Bornyl acetate (**88**)	-	1286	
	α-Pinene (**40**)	3.95	-	
	Fenchone (**41**)	-	1089	
	Camphor (**89**)	18.14	-	
	Camphene (**90**)	8.86	-	
	Sabinene (**91**)	10.27	-	
	*trans*-α-Bergamotene (**92**)	7.73	-	
	β-Pinene (**93**)	10.36	-	
	α-Phellandrene (**46**)	11.94	-	
	α-Terpinene (**45**)	12.52	-	
	γ-Terpinene (**94**)	14.57	-	
	Terpinolene (**95**)	16.00	-	
	Estragole (**96**)	6.06	-	
	1–8-Cineole (**97**)	13.16	-	
	*p*-Cymene (**44**)	12.88	-	
	*cis*-β-Terpineol (**98**)	16.30	-	
Sesquiterpenoids	β-Copaene (**99**)	8.10	-	[56,57,58,59,60,62,63,65,66,67,68,69,70]
	α-Humulene (**57**)	29.26	-	
	*cis*-β-Farnesene (**100**)	29.98	-	
	β-Cubebene (**101**)	-	1394	
	α-Cadinene (**102**)	-	1537	
	Aromadendrene (**103**)	-	1529	
	α-Bisabolene (**104**)	-	1561	
	β-Bisabolene (**56**)	8.17	-	
	α-Bisabolol (**105**)	-	1642	
	Neoisolongifolene (**106**)	-	-	
	*trans-*β-Guaiene (**107**)	-	1499	
	*cis*-Muurola-3,5-diene (**108**)	8.31	1502	
	Nerolidol (**109**)	8.47	-	
	Bicyclogermacrene (**110**)	30.60	-	
	β-Elemene (**111**)	-	1387	
	β-Caryophyllene (**112**)	-	1419	
	δ-Cadinene (**113**)	31.45	-	
	Spathulenol (**114**)	32.96	-	
	α-Selinene (**115**)	30.85	-	
	Bicyclosesquiphellandrene (**116**)	29.05	-	
	α-Bergamotene (**117**)	28.79	-	
	Caryopyllene oxide (**118**)	-	1550	
	1,10-Di-epcubenol (**119**)	34.06	-	

RT = Retention Time; RI = Retention Index.

**Table 3 molecules-27-06350-t003:** The chemical constituents of *O. gratissimum*.

Class of Chemical Constituents	Chemical Constituents	RT (min)	Percentage (%)	References
Alcohols	Chlorogenic acid (**120**)	-	-	[83]
Fatty acids	Methyl acetate (**121**)	30.55	-	[84]
	Palmitic acid (**122**)	28.35	-	[84]
Flavonoids	Luteolin (**123**)	9.06	-	[47,84,85,86,87,88]
	Apigenin (**124**)	12.36	-	
	Quercetin (**125**)	-	-	
	Epicatechin (**126**)	-	-	
	Nevadensin (**127**)	18.45	-	
	Salvigenin (**128**)	25.13	-	
	Morin (**129**)	-	-	
	Xanthomicrol (**130**)	18.23	-	
	Apigenin dimethyl ether (**131**)	27.46	-	
Phenols	Eugenol (**33**)	-	74.83	[38]
	Methyl eugenol (**132**)	6.36	-	
Phenylpropanoids	Sinapic acid (**133**)	2.72	-	[36,89]
	Rosmarinic acid (**134**)	3.82	-	
	Nepetoidin A (**135**)	17.34	-	
Monoterpenoids	Methyl carvacrol (**136**)	-	1.19	[89,90,91,92]
	Carvacrol (**50**)	-	0.20–8.40	
	1–8-cineole (**97**)	-	0.30–23.04	
Sesquiterpenoids	*trans-*α-Bergamotene (**92**)	-	0.20–0.70	[83,84,86,90,91,92,93,94]
	δ-Cadinene (**113**)	-	0.30–3.00	
	β-selinene (**148**)	-	0.82–7.96	
	β-Caryophyllene (**112**)	-	0.39–7.23	
	Humulene (**57**)	-	4.40	
	β-Bergamotene (**149**)	-	1.03–2.29	
	δ-Cadinene (**113**)	-	0.30	
	γ-Cadinene (**150**)	-	0.61	
	Caryopyllene oxide (**118**)	-	0.50–3.02	
	τ-Cadinol (**151**)	-	3.5	
	α-Panasinsene (**152**)	-	-	
	β-Chamigrene (**153**)	-	1.61–2.84	
	β-Copaene (**99**)	-	0.27	
	Germacrene-D (**154**)	-	0.10–29.9	
	β-Bisabolol (**155**)	-	-	
	β-Cubebene (**101**)	-	0.08	
	*Epi*-Cubebol (**156**)	-	0.21	
	*Epi*-Cubenol (**157**)	-	0.23	
	Copaene (**158**)	-	0.30–7.20	
	β-Bourbonene (**159**)	-	0.21–0.89	
	β-Selinene (**160**)	-	0.85–7.96	
	Bicyclogermacrene (**110**)	-	0.40–2.90	
	Calamenene (**161**)	-	0.38	
	α-Thujone (**33**)	-	0.1	
	β-Myrcene (**37**)	-	0.14	
	Camphene (**90**)	-	0.10–0.60	
	Sabinene (**91**)	-	0.18–0.90	
	2-Carene (**137**)	-	-	
	Camphor (**89**)	-	0.10–0.60	
	β-Pinene (**93**)	-	0.11–2.22	
	*trans*-Longipinocarveol (**138**)	-	-	
	γ-Terpinene (**94**)	-	8.23–22.90	
	4-Carene (**139**)	-	-	
	β-(*E*)-Ocimene (**140**)	-	0.10–0.43	
	β-(*Z*)-Ocimene (**141**)	-	0.21–4.00	
	Allo-Ocimene (**142**)	-	-	
	4-Methylstyrene (**143**)	-	-	
	Styrene (**144**)	-	-	
	Limonene (**48**)	-	1.25–5.27	
	*p*-Cymene (**44**)	-	12.04–25.00	
	Terpinene-4-ol (**145**)	-	1.90–4.35	
	Borneol (**51**)	-	0.20–0.55	
	Umbellulone (**146**)	-	-	
	α-Terpineol (**147**)	-	0.10–0.92	
	Estragole (**96**)	-	0.20–1.50	
	Thymol (**29**)	-	8.50–46.99	
	α-Pinene (**40**)	-	0.24–2.66	
	α-Curcumene (**162**)	-	0.26	
	Isolongifolol (**163**)	-	0.10	
	Isopinocamphone (**164**)	-	0.40	
	Longifolene (**165**)	-	3.00	
	γ-Muuroline (**166**)	-	0.16–3.88	
	α-*p*-Dimethyl styrene (**167**)	-	1.19	
	β-Ylangene (**168**)	-	2.7	
Triterpenoids	Oleanolic acid (**169**)	33.60	-	
Stigmastanes	Basilimoside (**170**)	29.12	-	

(-) = Test not performed; RT = Retention Time; RI = Retention Index.

**Table 4 molecules-27-06350-t004:** The chemical constituents of *O. campechianum*.

Class of Chemical Constituents	Chemical Constituents	KI	Percentage (%)	References
Aliphatic alcohol	3-Octanol (**73**)	988	-	[99]
Fatty alcohol	1-Vinylhexanol (**77**)	974	-	[99]
	(3*Z*)-Hexenol (**171**)	850	-	[99]
Monocyclic ketone	Carvone (**13**)	1239	-	[99]
Benzenoids	Benzene acetaldehyde (**172**)	1036	-	[99]
Phenols	Eugenol (**33**)	-	9.00	[99]
	Methyl eugenol (**132**)	-	12.00	
Polymethoxyflavones	5-Demethylnobiletin (**173**)	-	-	[99,103]
	5-Demethylsinensetin (**174**)	-	-	
Organooxygen compounds	(2*E*)-Hexenal (**175**)	846	-	[99]
	3-Octanone (**176**)	979	-	
Carboxylic acid esters	(3*E*)-Hexenyl acetate (**177**)	1001	-	[104]
Monoterpenoids	δ-3-Carene (**178**)	1008	-	[99,104,105]
	α-Terpinene (**45**)	1014	-	
	*p*-Cymene (**44**)	1020	-	
	β-(*E*)-Ocimene (**139**)	1044	-	
	β-(*Z*)-Ocimene (**140**)	1032	-	
	Neoalloocimene (**179**)	1140	-	
	Terpinolene (**95**)	1086	-	
	Limonene (**48**)	-	0.30	
	α-Terpinol (**84**)	-	0.30	
	Estragole (**96**)	-	-	
	1,8-Cineole (**97**)	-	3.30	
	Isoborneol (**180**)	1155	-	
	(+)-4-Terpinol (**144**)	1174	-	
	Linalool (**36**)	-	2.90	
	Myrcene (**37**)	-	0.20	
	β-Pinene (**93**)	-	0.80	
	α-Thujene (**181**)	924	-	
	Tricyclene (**182**)	921	-	
	*Cis*-sabinen hydrate (**183**)	1065	-	
	α-Pinene (**40**)	-	0.20	
	Camphene (**90**)	-	0.40	
	trans-α-Bergamotene (**92**)	1432	-	
	Sabinene (**91**)	-	0.10	
	Camphor (**89**)	1141	-	
	Sesquisabinene (**184**)	-	0.20	
Sesquiterpenoids	Humulene epoxide II (**185**)	1608	-	[99,105]
	β-Bisabolene (**56**)	1505	-	
	Germacrene-D (**154**)	-	10.10	
	Bicyclogermacrene (**110**)	1500	-	
	Germacrene-B (**186**)	1559	-	
	α-Humulene (**57**)	-	2.80	
	Aromadendrene (**103**)	1439	-	
	α-Copaene (**158**)	-	1.90	
	β-Bourbonene (**159**)	1387	-	
	α-Gurjunene (**187**)	1409	-	
	Viridiflorol (**188**)	1592	-	
	β-Eudesmol (**189**)	1649	-	
	3,7(11)-Eudesmadiene (**190**)	-	-	
	α-Guaiene (**191**)	-	5.60	
	γ-Muurolene (**192**)	-	0.30	
	α-Bulnesene (**193**)	-	7.10	
	Spathulenol (**114**)	-	0.40	
	τ-Cadinol (**151**)	-	-	
	Junenol (**194**)	1618	-	
	β-elinene (**148**)	-	-	
	δ-Cadinene (**113**)	-	2.00	
	Caryophyllene oxide (**195**)	-	0.40	
	E-Caryophyllene (**112**)	-	4.05	
	β-Elemene (**111**)	-	0.53	
	γ-Elemene (**196**)	-	0.60	
	Bicycloelemene (**197**)	-	0.20	

(-) = Test not performed; KI = Kovat’s Index.

**Table 5 molecules-27-06350-t005:** The chemical constituents of *O. sanctum*.

Class of Chemical Constituents	Chemical Constituents	RT (min)	RI	References
Fatty acids	Ocimumnaphthanoic acid (**198**)	-	-	[119,120,121]
	Methyl 9-methyltetradecanoate (**199**)	-	-	
	Ethyl 13-methyl-tetradecanoate (**200**)	-	-	
	Methyl 7-*Z*-hexadecanoate (**201**)	-	-	
	Ethyl palmitate (**202**)	-	-	
	Ethyl isovalerate (**203**)	-	-	
Carboxylic acids	(*Z*)-3-Hexenil acetate (**204**)	-	1005	[122]
Aliphatic aldehyde	Citronellal (**17**)	-	-	[113]
Benzenoids	Protocatechuic acid (**205**)	-	-	[113,120]
	Di-*n*-2-propylpentylphthalate (**206**)	-	-	
Unsaturated hydrocarbons	1,2,4- Trimethylcyclohexane (**207**)	13.36	-	[113]
Organooxygen compounds	1,2-Cyclopentanedione (**208**)	-	-	[119,120]
	Citrusin C (**209**)	-	-	
Phenolic	Vanillin (**31**)	-	-	[113,115,123]
	Methylisoeugenol (**210**)	7.50	-	
	Vanillic acid (**211**)	-	-	
	Sinapic acid (**132**)	7.20	-	
	*p*-coumeric acid (**212**)	-	-	
	*p*-hydroxybenzoic acid (**213**)	-	-	
	Ferulic acid (**214**)	-	-	
	Eugenol (**33**)	-	1382	
	Bieugenol (**215**)	-	-	
Flavonoids	Galuteolin (**216**)	-	-	[113,119]
	Isovitexin (**217**)	-	-	
	Chrysoeriol (**218**)	-	-	
	Cirsilineol (**68**)	-	-	
	Isothymunin (**219**)	-	-	
	Isothymusin (**220**)	-	-	
	Cirsimaritin (**221**)	-	-	
	Molludistin (**222**)	-	-	
	Vitexin (**64**)	-	-	
	Orientin (**223**)	-	-	
	Isoorientin (**224**)	-	-	
	Vicenin (**62**)	-	-	
	Luteolin-5-glucoside (**225**)	-	-	
	Esculin (**226**)	-	-	
	Esculectin (**227**)	-	-	
Phenylpropanoids	Rosmarinic acid (**133**)	6.20	-	[115,124]
Monoterpenoids	Estragole (**96**)	-	1229	[113,115,119,120,121,124,125]
	Carvacrol (**50**)	-	-	
	Linalool (**36**)	-	1109	
	α-Terpinyl acetate (**228**)	-	-	
	Terpiniolene (**95**)	-	-	
	Bornyl acetate (**88**)	-	1289	
	α-Bergomatene (**117**)	-	1469	
Sesquiterpenoids	Spathulenol (**114**)	-	1578	
	α-Humulene (**57**)	-	1482	
	α-Guaiol (**229**)	-	-	
	2-Methylene-4,8,8-trimethyl-4-vinyl- bicyclo[5.2.0]nonane (**230**)	-	-	
Triterpenoids	Oleanolic acid (**169**)	-	-	
	Ursolic acid (**231**)	8.50	-	

(-) = Test not performed; RT = Retention Time; RI = Retention Index.

**Table 6 molecules-27-06350-t006:** Data of inhibition zone of *Ocimum* species.

Microorganism	Extract	Inhibition Zone (mm)					Positive Control (mm)	Reference
		1	2	3	4	5		
**Gram-Positive Bacteria**								
*S. mutans*	Essential oil	28	-	14	-	-	^1a^21	[126,127,128]
	Methanol	-	8.30–9.40	-	-	-	^2a^10.90	
	Ethyl acetate	-	-	-	-	24	^5b^29.40	
*L. casei*	Essential oil	19	-	-	-	-	-	[126]
*E. faecalis*	Essential oil	15.67	-	-	10	-	-	[41,129,130,131]
	Methanol	-	6.90–10.40	20–26	-	-	^2a^15.50; ^3c^20	
*E. faecium*	Essential oil	15.67	-	-	9	-	-	[41,131]
*S. aureus*	Essential oil	33.33	29.20–30.56	-	11.67	41.50	^2e^23.93	[41,128,131,132,133,134,135,136]
	70% hydroethanol	9.33–11.17	-	-	-	-	-	
	Hexane	-	-	-	-	2.36	-	
	Ethyl acetate	-	-	-	-	30	^5b^29.30	
	Ethanol	-	-	-	-	12	-	
*B. cereus*	Ethyl acetate	10	-	-	-	21	^5b^29.30	[25,128,132,136,137]
	70% hydroethanol	9.50–16.33	-	-	-	-	-	
	Essential oil	-	10.66–16.11	-	-	-	^2e^20.53	
	Chloroform	-	-	12	-	-	^3f^15	
*Clostradium penfrigens*	Ethyl acetate	10	-	-	-	-	-	[138]
*S. phyogenes*	Essential oil	-	19.00	-	-	-	^2g^11.00	[139,140]
	Ethanol	-	-	6–10	-	-	-	
*Cutibacterium acnes*	Essential oil	-	16.78–18.13	-	-	-	^2f^13.13	[141]
*Lactococcus garvieae*	Ethanol	-	1.90	-	-	-	^2i^20.00	[142]
*B. subtilis*	Methanol	-	13.58	-	-	-	^2d^22.01	[128,143]
	Ethyl acetate	-	-	-	-	26	^5b^28.70	
*S. sanguinis*	Methanol	-	7.80–11.40	-	-	-	^2a^17.90	[129]
*S. faecalis*	Ethanol	-	-	10	-	-	-	[144]
*L. monocytogenes*	Essential oil	12	10.60–11.70	-	8	-	^1j^33; ^2j^33.70	[92,131,138]
	Ethanol	-	-	8–26	-	-	-	
*L. ivanovii*	Essential oil	13.67	-	-	12.67	-	-	[41,131]
*S. epidermidis*	Essential oil	22.67	-	-	11	-	-	[41,131,135,145]
	Ethanol	-	-	-	-	9	-	
	Bark	-	15	-	-	-	^2k^22	
*M. luteus*	Essential oil	14	-	-	-	-	^1c^26	[30]
*C. perfringens*	Essential oil	-	-	-	-	17.50	^5c^41.39	[25]
*Lb. plantarum*	Essential oil	-	-	-	-	12.58	^5c^15.38	[138]
*S. agalactiae*	Ethyl acetate	-	-	-	-	20	^5b^30.40	[128]
**Gram-Negative Bacteria**								
*P. gingivalis*	Essential oil	30	-	-	-	-	-	[146]
*P. intermedia*	Essential oil	30	-	-	-	-	-	
*F. nucleatum*	Essential oil	24	-	-	-	-	-	
*P. vulgaris*	Essential oil	-	-	-	11.33		-	[41,131]
*E. coli*	Essential oil	10.67	17.48–23.58	12	11.67	15.40	-	[41,128,131,132,133,134,147]
	Ethyl acetate	9	-	-	-	15.70	^5b^31.30	
	Hexane	-	-	-	-	2.26	-	[25]
*Klebsiella pneumonia*	Ethyl acetate	10	-	-	-	15.30	^5b^29.30	[128,148]
	Methanol	-	-	16–23	-	-	-	
*Salmonella paratyphi*	Ethyl acetate	9	-	-	-	-	-	[25]
*V. chlorea*	Essential oil	18	-	-	-	-	-	[30]
*S. typhimirium*	Essential oil	13.00–20.10	20.00	-	-	-	^1j^28.80; ^2h^11.00	[92,128,139]
	Ethyl acetate	-	-	-	-	16.70	^5b^29.70	
*S. dysenteria*	Essential oil	18	-	-	-	-	^1c^17	[30,145]
	Bark	-	16	-	-	-	^2k^25	
*P. aeruginosa*	Essential oil	-	Max	-	-	12	^2e^12.64	[128,132,135,149,150]
	Leaves	-	-	16	-	-	^3k^39	
	Ethyl acetate	-	-	-	-	10.30	^5b^22.30	
	Ethanol	-	-	-	-	11	-	
*A. baumannii*	Essential oil	-	15.30	-	-	-	-	[151]
*Salmonella enteritidis*	Essential oil	-	18.00	-	-	-	-	[152]
*S. typhi*	Essential oil	-	24.80	-	-	-	-	[136,153]
	Chloroform	-	-	14	-	-	^3c^16	
*P. multocida*	Essential oil	-	13.60–18.40	-	-	-	^2h^31.10	[154]
*Listonella anguillarum*	Ethanol	-	3.90	-	-	-	^2i^20.00	[142]
*Yersinia ruckeri*	Ethanol	-	1.50	-	-	-	^2i^20.80	[142]
*P. mirabilis*	Leaves	-	-	11–22	-	-	^3d^15 mm; ^3l^29 mm	[128,148,155]
	Methanol	-	-	25–32	-	-	-	
	Ethyl acetate	-	-	-	-	15.10	^5b^30.30	
*Shigella gexineri*	Essential oil	-	-	16	-	-	^3m^29	[156]
*Proteus mirabilis*	Flavonoids	-	-	12–28	-	-	^3e^24	[148]
*Shigalla* sp.	Ethanol	-	-	8	-	-	-	
*Salmonella* sp.	Ethanol	-	-	6	-	-	-	
*Shigella dysenterae*	Chloroform	-	-	2	-	-	^3n^14	[136]
*P. fluorescens*	Essential oil	-	-	-	-	20	^5m^12.33	[135]
*Shigella boydii*	Ethyl acetate	-	-	-	-	14.30	^5b^30.70	[128]
*Yersinia enterocolitica*	Ethyl acetate	-	-	-	-	20.10	^5b^28.30	
*Aeromonas hydrophila*	Ethanol	-	-	-	-	15	-	[135]
*Vibrio harveyi*	Ethanol	-	-	-	-	12	-	

(-) = Test not performed; 1 = O. americanum, 2 = O. basilicum, 3 = O. gratissimum, 4 = O. campechianum, 5 = O. sanctum. Positive control: ^a^(Chlorhexidine), ^b^(Ciprofloxacin), ^c^(Gentamicin), ^d^(Ciprofloxacin), ^e^(Tetracycline), ^f^(Clindamycin), ^g^(Rifampicin), ^h^(Amoxicilin), ^i^(Oxytetracycline), ^j^(Ampicillin), ^k^(Streptomycin), ^l^(Ketoconazole), ^m^(Penicillin), ^n^(Cloxacillin).

## Data Availability

The study did not report any data.

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
