# Peer review of "Ocimum Species: A Review on Chemical Constituents and Antibacterial Activity"

_molecules, 2022, doi:10.3390/molecules27196350_

Round 1
Reviewer 1 Report
The manuscript entitled "Ocimum species: A Review on Chemical Constituents and Antibacterial Activity" by Dharsono et al. presents comparison of components of five types Ocimum species and their antibacterial activity.
The manuscript is poorly written with confusing sentences and should be completely revised. Authors should consider the following comments:
1. The tables containing the names of chemical compounds present in Ocimum are too long. Authors should present them in a concise and clear way.
2. Authors should analyse tables 1, 3, 5, 7, 9 and correct the names of compounds e.g. "1-methyl pentyl", etc. Used nomenclature should be unified and consistent with IUPAC rules.
3. The splitting of components into chemical groups also needs correction e.g. octyl acetate and methyl acetate do not belong to carboxylic acids etc.
4. Figures 2, 4, 6, 8 and 10 should be moved to Supplementary.
5. The data included in tables with antibacterial activity for all species should be compared and discussed. Authors should pay particular attention to the similarity and differences of individual species. A comprehensive discussion is required in order to highlight the achievements presented in the presented manuscript.
Author Response
In order to make it easier for reviewers to check the revised section, we mark it with a green highlighter.
- The tables containing the names of chemical compounds present in Ocimum are too long. Authors should present them in a concise and clear way.
- Authors should analyse tables 1, 3, 5, 7, 9 and correct the names of compounds e.g. "1-methyl pentyl", etc. Used nomenclature should be unified and consistent with IUPAC rules.
Answer:
Thank you for your correction.
For the comments number 1 and 2, authors have edited for the name of chemical compounds present in Ocimum, revised them to become unified and consistent.
- The splitting of components into chemical groups also needs correction e.g. octyl acetate and methyl acetate do not belong to carboxylic acids etc.
Answer:
Thank you for your correction.
Author have revised some compounds into the correct chemical group, e.g octyl acetate is moved to fatty alcohols.
- Figures 2, 4, 6, 8 and 10 should be moved to Supplementary.
Answer:
Thank you for the advice given.
We have also moved the figure of chemical components into supplementary files.
- The data included in tables with antibacterial activity for all species should be compared and discussed. Authors should pay particular attention to the similarity and differences of individual species. A comprehensive discussion is required in order to highlight the achievements presented in the presented manuscript.
Answer:
Thank you for the advice given.
We have revised to take the antibacterial activity data of Ocimum species into one table, so that we can compared the activities of five Ocimum species.
Reviewer 2 Report
Thank you for your efforts in preparation of such a review article. I have several comments as follows:
- Some compounds in the tables are better moved to other groups for example, palmitoleic acid it is a fatty acid, why it has been assigned to a different group. Also what is meant by nitro, its structure should be corrected.
- It is better to draw the structures with their stereochemistry.
- I am still worried about adding photos of the plants even cited, I think you have to get a permission. Please check. I am not totally sure of that.
- A relative comparison of the biological activity of the five species should be discussed with an emphasis on the difference in chemical composition, if any.
- Also If I would like to make use of your study, how can I consume Ocimum to get its antibacterial activity? Please discuss in details.
Author Response
In order to make it easier for reviewers to check the revised section, we mark it with a yellow highlighter.
- Some compounds in the tables are better moved to other groups for example, palmitoleic acid it is a fatty acid, why it has been assigned to a different group. Also what is meant by nitro, its structure should be corrected.
Answer:
Thank you for the correction.
Authors have edited some compounds into the correct chemical group, and we have also edited the structure of nitrocyclohexane.
- It is better to draw the structures with their stereochemistry.
Answer:
Thank you for the advice given.
Authors have revised the figure of chemical components by adding the stereochemistry of them. We moved it to supplementary files because of a suggestion from reviewer 1.
- I am still worried about adding photos of the plants even cited, I think you have to get a permission. Please check. I am not totally sure of that.
Answer:
Thank you for the comments
We consider adding an image with a reference to be sufficient, but if a permission is needed, we could not fulfill it so we prefer to remove it.
- A relative comparison of the biological activity of the five species should be discussed with an emphasis on the difference in chemical composition, if any.
Answer:
Thank you for the advice given.
We have added the GC-MS data of the chemical components to give it the scientific value, but the GC-MS data of some chemical compounds have not been determined, so we couldn’t add them in manuscript.
- Also If I would like to make use of your study, how can I consume Ocimum to get its antibacterial activity? Please discuss in details.
Answer:
Thank you for the comment.
The preparation of Ocimum plants to be consumed as antibacterial agents certainly requires a long process. Utilization of this plant for consumption can be done in traditional ways such as by boiling Ocimum leaves and so on. We have added this description to the manuscript on lines 263-271.
Author Response
In order to make it easier for reviewers to check the revised section, we mark it with a red highlighter.
- Phytochemicals in Human Health Ocimum Phytochemicals and Their Potential Impact on Human Health” pp.1-26; Intech Open, DOI:http://dx.doi.org/10.5772/intechopen.8855.
Answer:
Thank you for the advice given.
Authors have added the citation of bhattacharjya et al. (2019) as suggested by reviewer in line 228-231.
- In a Review, a careful analysis of the selected publications must be considered and subsequently a discussion must be carried out. This basic methodology, was not done in this article. A careful analysis of the articles cited in the review was not performed. Several references are of low scientific quality. They contain errors in the methodology, that are not currently accepted in prestigious Journals. Those reported results are incorrect and were included in the review.
Answer:
Thank you for the advice given.
We have removed some reference which has low scientific quality, e.g “Sonia, A.; Reena Saad, F.; Mahmuda, S. Bacterial isolates and antimicrobial susceptibility in children with acute diarrhea at ibn sina medical college, bangladesh. Jurnal Kedokteran dan Kesehatan Indonesia 2017, 8, 80-86.”
- In the references, the phytochemical tests used for the identification of the different families of compounds are not acceptable. They are known to give false positive results. In relation to some articles where the composition of essential oils is reported, these results are incorrect, the RI (retention indexes) were not calculated and they were not compared with those published. Characterization was done using the GC/MS database, which is incorrect. Those data have no scientific value and reproducing them in a Review is dangerous for the science. See as F examples, references: 28, 29, 54, 156, etc.
Answer:
Thank you for the correction.
We have added the GC-MS data of the chemical components to give it the scientific value, but the GC-MS data of some chemical compounds have not been determined, so we couldn’t add them in manuscript.
- Antimicrobial resistance in Escherichia coli has increased worldwide and its susceptibility patterns show substantial geographic, population and environmental variations. Reference 1, only contemplates a case in children in one hospital and does not show the problem worldwide.
Answer:
Thank you for the comments.
It is a bit difficult to determine the problem caused by E. coli worldwide, but to strengthen our argument we revised the data on E. coli infections with more accurate data such as the percentage of antibiotic resistance, and tests performed in 52 hospitals in North America during 7 years.
- References 16-18 has nothing to do with the previous paragraphs.
Answer:
Thank you for the comments.
Authors try to check the reference number 16-18 which are Joshie et al. (2009), Kačániová et al. (2022), and Abd-Alla et al. (2013). These has been added in line 56-62. This references have discussed the content of secondary metabolites and essential oil components that can inhibit the growth of microorganisms. Then, in our opinion, this sentence in the manucript is related to the previous sentence which discusses the benefits of the Ocimum plant that has been used so far. These benefits come from the content of secondary metabolites found in this plant. So that the sentences written in the paragraph are related to each other.
Reviewer 4 Report
The review manuscript presents 5 species of Ocimum, with their main constituents and their antibacterial activity.
Major Revisions
- Although the chemical constituents of each of the species have been presented, no quantitative data is presented for each compound or group of compounds. This lack of information makes it difficult to compare the chemical composition between these species or other species reported in other manuscripts. This needs to be included.
- The discussion presented in lines 250 to 266 needs to be revised.
Considering the hydrophobic essential oil, it has a greater affinity for the outer layer of the cell wall of gram-negative bacteria, facilitating its penetration and damage to the plasma membrane. So, shouldn't its effect be greater on gram-negative bacteria?
- Although data on antibacterial activity have been compiled and presented for each species of Ocimum, there was a lack of a comparative table between species, using the same microorganism as a model, and mainly a discussion about which species has the highest activity and which main compounds contribute to this greater antibacterial activity.
This comparison may be the differential of this manuscript in relation to others already existing in the literature.
Minor Revisions
- Line 33: Correct Klebsiellap neumoniae,
- Line 69: O. americanum must be in italics.
- Standardize gram-positive or gram-negative (hyphen).
- What does the symbol (–) mean in antibacterial activity tables? No action or test not performed? inform in the tables.
Author Response
In order to make it easier for reviewers to check the revised section, we mark it with a blue highlighter.
Major Revisions
- Although the chemical constituents of each of the species have been presented, no quantitative data is presented for each compound or group of compounds. This lack of information makes it difficult to compare the chemical composition between these species or other species reported in other manuscripts. This needs to be included.
Answer:
Thank you for the correction.
Authors have added the quantative data to compare the chemical constituents, but the GC-MS data of some chemical compounds have not been determined, so we couldn’t add them in manuscript.
- The discussion presented in lines 250 to 266 needs to be revised.
Considering the hydrophobic essential oil, it has a greater affinity for the outer layer of the cell wall of Gram-negative bacteria, facilitating its penetration and damage to the plasma membrane. So, shouldn't its effect be greater on gram-negative bacteria?
Answer:
Thank you for the comments.
Basically, essential oils are able to inhibit both Gram-positive and negative bacteria by penetrating the cells in the cell wall and cytoplasm of the bacteria. However, as explained in lines 235-237, there are differences in cell wall structure, where Gram-negative bacteria have a more complex structure so that they will be more resistant to essential oils. To strengthen this argument, we also add related references as in lines 238-244.
- Although data on antibacterial activity have been compiled and presented for each species of Ocimum, there was a lack of a comparative table between species, using the same microorganism as a model, and mainly a discussion about which species has the highest activity and which main compounds contribute to this greater antibacterial activity. This comparison may be the differential of this manuscript in relation to others already existing in the literature.
Answer:
Thank you for the advice given.
We have revised the table of antibacterial activities to take them into one table and have discussed the greater antibacterial activity.
Minor Revisions
- Line 33: Correct Klebsiellap neumoniae,
- Line 69: O. americanum must be in italics.
- Standardize gram-positive or gram-negative (hyphen).
- What does the symbol (–) mean in antibacterial activity tables? No action or test not performed? inform in the tables.
Answer:
Thank you for the correction.
We have also revised the incorrect word in line 33 and 69, then standardize ‘gram-positive’ and ‘gram-negative’, and annotate the symbols ‘-‘.
Round 2
Reviewer 3 Report
Suggested modifications were made as requested